# Implementation and enforcement of mandatory calorie labelling regulations for the out-of-home sector in England: Qualitative study of the experiences of business implementers and regulatory enforcers

**Michael Essman**[1]*, **Tom Bishop**[1], **Thomas Burgoine**[1], **Andrew Jones**[2], **Megan Polden**[3], **Eric Robinson**[3], **Richard Smith**[4], **Jean Adams**[1], **Martin White**[1]

1 MRC Epidemiology Unit, University of Cambridge, Cambridge, Cambridgeshire, United Kingdom,
2 School of Psychology, Liverpool John Moore's University, Liverpool, Merseyside, United Kingdom,
3 Institute of Population Health Sciences, University of Liverpool, Liverpool, Merseyside, United Kingdom,
4 Department of Public Health and Sports Science, University of Exeter, Exeter, Devon, United Kingdom

* mike.essman@mrc-epid.cam.ac.uk

**Data availability statement:** Data supporting this study cannot be made available due to ethical restrictions imposed by the University of Cambridge Humanities and Social Sciences

## Abstract

### Background

Mandatory calorie labelling on menus of large out-of-home food outlets was implemented in England on 6 April 2022. Barriers and facilitators that were unforeseen before implementation may modify policy impacts. As part of a process evaluation, we aimed to examine the implementation of calorie labeling in England, focusing on business experiences and local authority enforcement to identify barriers and facilitators to achieving policy goals.

### Methods

Using purposive sampling, we recruited 11 employees of large food businesses (implementers) and 9 employees of LA environmental health or trading standards departments (enforcers). Post-implementation semi-structured interviews were conducted by video conference. Interviews were audio recorded, transcribed verbatim and analysed using the Framework Method.

### Results

Both groups of participants described a decentralised approach to delivery and enforcement, and resource constraints meant LAs were unable to assist with all business inquiries. Enforcement activity was limited because complaints about labelling from the public were rare, and enforcers prioritized acute food safety issues. Pre-implementation discussions created the presumption among enforcers that most businesses were compliant. Implementers claimed that businesses wanted to

Research Ethics Committee (HSS REC), as the interview transcripts contain potentially identifiable and sensitive information. Participants did not consent to public sharing of their data, and sharing even anonymized transcripts risks compromising participant confidentiality. Queries regarding data can be made to Data Sharing at the University of Cambridge MRC Epidemiology Unit at the following email address: datasharing@mrc-epid.cam.ac.uk. Queries regarding ethical restrictions can be made to the University of Cambridge HSS REC at the following email address: HSSREC@admin.cam.ac.uk.

**Funding:** This independent research was commissioned and funded by the NIHR (NIHR200689, Policy Research Programme). The views expressed in this publication are those of the author(s) and not necessarily those of the NIHR or Department of Health and Social Care. This work was supported by the MRC Epidemiology Unit, University of Cambridge [grant number MC/UU/00006/7]. Funders had no role in study design; collection, analysis and interpretation of data; writing the report; and the decision to submit the report for publication.

**Competing interests:** The authors have declared that no competing interests exist.

comply to safeguard their reputation and maintain customer trust. While participants supported calorie labelling, potential barriers to policy impact included a presumed lack of customer interest. Financial pressure during implementation strained business resources, and businesses suggested that customers may prioritise financial over health concerns in their decision-making.

## Conclusions

These findings underscore the need for central guidance, verification of adherence, and sufficient enforcement resources. To optimize policy success, future developments should consider economic contexts, customer expectations, and policy refinement, while recognizing common industry arguments against policy implementation.

## Introduction

Eating food from out-of-home food outlets (OHFO) is associated with poorer dietary quality, increased caloric intake and weight gain [1–4]. The out-of-home food sector includes physical and online locations where food and beverages are sold for immediate consumption, such as restaurants, cafés, pubs and bars, takeaways, fast food, street food, and other sites [5]. A recent study in the UK found that over 90% of meals analysed from fast food and full-service restaurants exceeded national recommendations on per meal calorie content (600kcal), and approximately half of the meals contained over 1000 kcal [6]. Customer intercept surveys involving over 3,300 participants in England indicated a substantial underestimation of purchased calories by an average of 253 kcal, with over two-thirds of customers choosing meals exceeding 600 kcal [7]. As a policy response to inform customers about the calorie contents of menu items and reduce calories consumed out-of-home, mandatory calorie labelling laws have been implemented in national and subnational jurisdictions [8].

Mandatory calorie labelling in large OHFO was implemented in England on the 6th April 2022. Food businesses are in scope of the policy if they sell food in a form for immediate consumption that is not pre-packaged, and the business has at least 250 employees [9]. Exempt establishments include education institutions for pupils <18 years; workplace canteens solely used by employees; and health and social care settings where food is solely provided for patients or residents. Specific item exemptions include menu items available for less than 30 days, beverages with greater than 1.2% alcohol content by volume, loose fruit and vegetables and condiments added by customers. Written calorie labels, alongside a reference statement that adults need around 2000 kcal a day, are required for both online and in store purchases at all points of choice, defined as any place where customers choose what food to buy. Businesses can use laboratory analysis or estimations based on food composition databases and ingredients to determine calorie content for labels [9].

Two aims of the calorie labelling legislation are to encourage customers to make informed (and thus healthier) decisions, and to encourage businesses to reformulate their products to lower calorie offerings [9]. However, the evidence for the impact

of calorie labelling on reformulation and customer choices is mixed. A meta-analysis of lab-based experimental studies, cross-sectional studies, difference-in-difference, pre-post observational studies, and pre-post studies with controls found that calorie labelling interventions were associated with 15 fewer kcal per item sold by businesses and 21 fewer kcal ordered per customer [10]. More recent studies from the USA found no significant reformulation but an introduction of new lower calorie menu times in large chain restaurants and small decreases in mean calorie and nutrient content of fast-food meals after calorie menu labelling [11,12]. Regarding evidence for changing customer behaviour, some reviews have found small potential reductions in calories consumed [10,13]. Studies implemented in real world (not laboratory) settings with control groups have identified limited evidence of the effect of calorie labels on customer choices [14]. A more recent review of national, state, and municipal menu labelling policies found that most evidence for effectiveness came from observational and longitudinal studies in the United States when lower calorie items were introduced on menus, but there was limited evidence for effectiveness for case-control and quasi-experimental studies [8].

This study is part of a process evaluation on the impact of the calorie labelling policy in England. This comprehensive impact assessment includes examination of OHFO compliance with the regulations [15], pre-post changes in customer purchases [16], pre-post changes in customer behaviours associated with menu labelling [17], and pre-post changes in the energy content of OHFO menu items [18]. Customer surveys examining pre-post changes in customer purchases indicate that, despite moderate adherence, implementation of the English calorie labelling laws is not associated with a change in calories purchased or consumed [16]. Post-implementation compliance checks at large food businesses found 80% of outlets surveyed implemented any form of calorie labelling, 67% of outlets surveyed had legible calorie labelling text, and 15% of outlets met all implementation criteria [15], Thus, imperfect implementation of calorie labelling, and minimal evidence for reformulation could contribute to the lack of change in calories purchased. This study sought to understand the implementation process and to identify causes of imperfect implementation.

As far as we are aware, no previous research has explored barriers and facilitators to successful implementation of calorie labelling policies in England or elsewhere. Therefore, the aims of this study were to examine the experiences and processes of implementing calorie labelling in England from the perspectives of businesses and local authority enforcement, identifying barriers, facilitators, and contextual factors that influence policy effectiveness. We were also open to identifying unforeseen themes that emerged from interviews.

## Methods

This study is reported as per the consolidated criteria for reporting qualitative research (COREQ) guidelines (S1 File) [19]. We completed one-to-one, semi-structured qualitative interviews with employees of large food businesses (implementers of the regulations) and employees of local authorities (LAs)—local government bodies—who were enforcers of the regulations. There are 317 LAs in England, with a mean population size of approximately 178,000 per LA. The study was reviewed by and received ethical approval from the Humanities and Social Sciences Research Ethics Committee at the University of Cambridge: reference 22.294.

### Recruitment

To recruit implementers, we used purposive sampling to achieve variation in business types that have been used in prior research on the out-of-home food sector in England including fast food, cafes, restaurants, and pubs [20]. The inclusion criteria for implementers were [1] being employed by a food business subject to the calorie labelling regulations implemented in England in April 2022 [21], and [2] being involved in the delivery of the menu labelling. Being involved included either or both at the strategic level (how to respond to the new regulations), and the operational level (how labels should appear on and be added to menus, and how calorie values should be calculated). We also included representatives of trade organisations because these organisations often represented food business interests to the Government. Initial contacts were sent to employees at 22 food businesses or organizations, from which 12 agreed to interview. One person

dropped out before the interview citing their manager no longer wanted them to conduct the interview. No more than one participant was recruited from each organisation. Contact details for interviewees were sourced via business websites and LinkedIn, a professional networking platform [22]. Before recruitment began, we aimed to recruit up to a maximum of 25 implementers. Recruitment was stopped after reaching thematic saturation, which was operationalized as no new themes related to implementation or enforcement provided in 5 consecutive interviews [23]. Field notes assisted with reflections during interviews and helping to identify repeating themes.

To recruit enforcers, we used purposive sampling of LAs to represent four geographical areas (North, Midlands, South and London) and all five quintiles of income deprivation according to the Office for National Statistics [24]. To cover the different types of enforcement activities, we included LAs with and without Primary Authority relationships with large food businesses. The Primary Authority is an LA relationship with business wherein businesses "receive assured and tailored advice on meeting environmental health [or] trading standards" [25]. The inclusion criteria for enforcers were working for an LA and active involvement in enforcing the regulations or providing guidance to businesses as part of a Primary Authority agreement. Thirty-six LAs were initially contacted, from which 10 agreed to interview. One interviewee dropped out before the interview, citing that they had nothing relevant to policy enforcement to report. No more than one participant was recruited from each organisation. Contact details for interviewees were sourced via local government websites and LinkedIn. Before recruitment began, we aimed to recruit up to a maximum of 15 enforcers. Recruitment was stopped after reaching thematic saturation, which was operationalized as no new themes related to implementation or enforcement provided in 5 consecutive interviews [23].

An initial introductory template email was sent to potential interviewees, followed by an approximately 15-minute telephone call to explain the study purpose and check eligibility. All contacts were also sent a Participant Information Sheet (S2 File) explaining the study and its purpose. After the introductory call, if the potential participants met the inclusion criteria and were willing to proceed, then they signed a written electronic consent form prior to the recorded interview.

## Data collection: interviews

Semi-structured interviews directed by topic guides were conducted by ME using video conferencing software, and participants were told the interview could last up to 60 minutes. Interviews were conducted from 28 June 2022 to 15 December 2022. Topic guides for implementers and enforcers were similar in the broad topics covered, such as the feasibility of implementation/enforcement, potential barriers and challenges, and the expected impacts of the policy on businesses/local authorities and customers. However, they were specifically adapted to address the unique perspectives and experiences of each group. The guide for implementers focused on their internal processes, including how they implemented the labelling requirements, challenges faced, and changes in business practices. The guide for enforcers was designed to explore regulatory aspects, such as enforcement challenges, compliance monitoring, and interactions with businesses. This tailored approach ensured that we captured a range of experiences related to both implementing and enforcing the calorie labelling policy.

The interview topic guides were developed collaboratively by the research team to address the aims of the work. They were piloted in two instances: first, with a colleague who has led numerous policy-relevant qualitative interview research, and second, in a presentation to the departmental qualitative methods group. The full interview topic guides are provided as supporting files S3 & S4 File Interviews were digitally audio recorded and transcribed verbatim by a trusted external transcription company. Transcripts were checked for accuracy and anonymised for analysis.

## Data coding and analysis

Given the pragmatic and descriptive aims of this study, which sought to evaluate and inform policy rather than develop or test theoretical constructs, we used the Framework Method for data analysis [26]. Employing a broadly deductive approach, pre-specified topic guides were used to answer questions related to the experiences and processes of

implementing calorie labelling in addition to a more inductive approach to generate open, unrestricted codes from the interview data (S3 & S4 File). Three researchers (ME, JA, MW) independently coded three transcripts to ensure important aspects of the data were not missed, and ME coded all remaining transcripts. The Constant Comparative Method and Deviant Case Analysis were used to ensure reliability of themes. We attempted to reduce our influence on participants' responses by following a standardised topic guide, asking for clarification where necessary, with the goal of capturing interviewees' perspectives and experiences as independent from our own expectations. The independent coding by three researchers with mixed methods experience was another procedure to reduce bias. Data were compared within and across participant interviews to identify key themes as well as contradictions or points of tension. During the data analysis phase, researchers most directly involved in analysis (ME, JA, MA) met with other members of the research team to share emerging insights from the data and seek alternative interpretations.

After codes were developed iteratively from the topic guides and transcripts, we proceeded with data analysis. Digital tools, including Microsoft Excel [27] and NVivo, version 12 [28], were used to develop analytical themes, and whiteboard diagramming was used to generate insights regarding the structural relationships between interviewees and themes encoded from the data. To ensure the research included insights from our entire study sample, data were charted into a matrix to compare cases and codes. Memos were developed for all themes and provided a substantive basis for reporting in the Results section. Anonymized verbatim quotations were used to illustrate findings [26].

### Reflexivity statement

This work was conducted by an inter-disciplinary team of academic researchers with expertise in dietary public health, evaluation of public health interventions, behavioural science, health geography, data science, health economics and mixed-methods research. ME, who is a male research associate with a PhD in nutrition with a minor in epidemiology and an MSc in Medical Anthropology, conducted the interviews. He completed training from the Social Research Association for planning and designing a qualitative study, and had meetings with the project lead, MW, who has extensive experience publishing qualitative research. We have conducted research on current UK implementation of calorie labelling, OHFO and the food they serve, and have experience of policy evaluations including of school food standards, television food advertising restrictions, the Soft Drinks Industry Levy and other soda taxes, and supermarket checkout food policies.

### Results

Our sample included 11 employees from different types of out-of-home food businesses and organisations (implementers of the regulations) including heads, directors and managers of product, policy, technical services and nutrition, and 9 employees of LAs from environmental health or trading standards (enforcers of the regulations) including principal environmental health and trading standards officers and relevant team leads (Table 1). Interviews revealed interdependent and nested themes that together may explain experiences and success, or otherwise, of the policy. The five themes encompass three contexts: the economic context, shaping both business and customer finances; a business context with pre-existing assumptions about customer behaviour at OHFOs; and a regulatory context involving the relationships between central government, local authorities, and businesses. Additionally, the themes cover potential downstream business impacts, focusing on the resources and capabilities required for calorie labelling policy implementation. A final theme is how the upstream contexts may influence customer behaviour (Table 2). We discuss each of these themes in turn below.

### Economic context

Participants reported that when the calorie labelling regulations were implemented, businesses were already struggling because of global financial challenges. Supply chain issues (shortages or higher costs) resulting from forced business

**Table 1. Business types represented by implementers and geographic regions represented by enforcers.**

| Implementers | |
|---|---|
| **Study ID** | **Organisation Type** |
| Implementer 01 | Café chain |
| Implementer 02 | Trade organisation |
| Implementer 03 | Café chain |
| Implementer 04 | Café chain |
| Implementer 05 | Pub chain |
| Implementer 06 | Group of chains |
| Implementer 07 | Pizza chain |
| Implementer 08 | Large catering business |
| Implementer 09 | Pub chain |
| Implementer 10 | Fast food chain |
| Implementer 11 | Trade organisation |
| **Enforcers** | |
| **Study ID** | **English Region** |
| Enforcer 01 | West Midlands |
| Enforcer 02 | Yorkshire and Humber |
| Enforcer 03 | South East |
| Enforcer 04 | South East |
| Enforcer 05 | South East (Greater London) |
| Enforcer 06 | North East |
| Enforcer 07 | East of England |
| Enforcer 08 | South West |
| Enforcer 09 | East of England |

closures due to COVID-19 and the economic fallout from the war in Ukraine were particularly mentioned. These conditions created additional challenges to implementing the calorie labelling regulations including frequent changes in ingredient supply.

Participants also felt that this economic background impacted customers. Businesses perceived that customers' concerns about long-term health and diet-related diseases had become subordinate to more immediate economic concerns, with the 'cost-of-living crisis' leading to a focus on 'value for money' when eating out of home. Paradoxically, therefore, calorie labelling may have facilitated making selections based on cost per calorie (Table 2).

### Business context of the out-of-home food sector

The out-of-home food business context included implementers' perceptions about why customers use the out-of-home sector and their expectations of the food served; implementers' claims about central government's understanding of their sector; and how calorie labelling fit into pre-existing business plans.

### Out-of-home food is a treat

According to many implementers, customers believe out-of-home eating is spontaneous and indulgent. One business that conducted detailed customer research summarized their findings, "*healthy is more planned, indulgence is spontaneous…*" (Implementer 09). This led some implementers to suggest a policy encouraging healthier out-of-home eating misunderstands how customers interact with the sector. Implementers suggested that even if customers are trying to restrict

Table 2. Descriptions of analytic themes, including descriptions, key quotes and implications for policy.

| Themes | Description of context | Key quotes |
|---|---|---|
| Economic context | Businesses: Due to the global financial situation, businesses were already facing challenges related to finances and supply chains. Adding regulations increased the challenges.<br>Customers: When eating out-of-home, customers may seek value for money because of the cost-of-living crisis. Reducing calories may not fit their goals and concerns. | "One thing that went wrong there was timing… the actual volume of it was very challenging… you had an issue where lots of people were on furlough, so nutritionists within businesses who normally would be looking after this area just weren't there… we did say, "can you just give us a bit more time to implement this". Not because we don't want to do it, we want to do it properly, but it's crisis mode and you've put something else in there." (Implementer 02)<br>"with the cost-of-living crisis people aren't bothered about calories, they want full bellies… They want to be able to feed their kids and they want things that are value for money." (Enforcer 02) |
| Business Context | OHFO eating is a treat – customers go for a treat, not to search for the healthiest options; calorie labelling may not match the purpose of eating out; businesses claim OHFOs are highly diverse; businesses want a level playing field that ensures if they must comply, then so must all competitors | "At the end of the day…dining out for people always was and most definitely at the moment is a treat, …even if they are actively trying to manage their calorie intake every other meal, when they eat out they will throw caution to the wind and eat what they want" (Implementer 05).<br>"I know the Government hate us saying this and it's not a way to try and get out of anything at all, but when consumers come into the out-of-home sector, they assume it is a treat. So, it doesn't impact them, they're not looking [at calories]" (Implementer 04). |
| Regulatory Context | Structure of enforcement and enforcement actions – Decentralized approach to enforcement, Primary Authority works with businesses upstream of regulations; calorie labelling lower priority for enforcement than acute threats to health such as allergens | "Our work is Primary Authority work which means that the businesses are very open and honest with us... we'd rather that they came to us and said we're not ready or we haven't done it or this has happened." (Enforcer 04).<br>"I think there is a reasonable level of compliance, which is probably reflected by the fact that we're not getting many complaints from people." (Enforcer 03) |
| Business Impacts | The degree to which calorie labelling regulations cost the organization money or other resources; aspects of policy implementation: how accurately calorie values are calculated, how menus changed including reformulated items, how the timeline affected feasibility | "the Government and what they estimated how much it was going to cost businesses was wildly underestimated. It has been a huge cost to get this rolled out... that tech piece has been so expensive and also so resource heavy" (Implementer 04)<br>"because we would've always got our menu boards printed anyway, I don't think it's any extra cost to add a few calories onto it... the only thing it does do is makes us have to print things a lot sooner in advance, also the pressure to get it right is greater" (Implementer 03) |
| Consumer Impacts | How customers respond: have they changed their purchases, complained about regulations or lack thereof, any other feedback to businesses/LAs | "I'm not sure whether [using calories to make healthier choices] is understood by everybody... There's only, at the moment, only a reference statement, statement of daily calorie needs to say adults need around 2,000 calories. So, there isn't a huge amount of education leading up to it or after the regulation was implemented." (Implementer 08)<br>"I think it would be better to push more understanding to the consumer of well yes, it might be this many calories but let's take a look in what is there... is it over-simplifying it so that people really are missing the point? So, I don't know about whether it is achieving its aim of reducing obesity." (Enforcer 04) |

calories, they may not do so when eating out-of-home (Table 2). Some implementers suggested that even very large meals were not important in the context of longer time scales.

*From a long-term health perspective... if you think about 2,000 calories a day, 14,000 calories a week, one restaurant meal per week is not a disproportionate amount. There's so many other inputs to that calorie build during the period of the week that dining out once a week makes virtually zero difference to your overall calorie consumption in a week or a month (Implementer 06).*

In many cases, implementers perceived that their food is not identified by customers as health promoting, but that is part of the attraction.

*They know what they're buying…we're a fun brand, people come for us for a treat, and as a treat it's going to be calorific on the whole, so generally we want to support the government strategies, but we also want to ensure that we're not alienating our loyal customer base (Implementer 07).*

## Diversity of the sector

The common claim that the out-of-home sector is an indulgent treat for customers contrasted with claims that the diversity of the out-of-home sector was not reflected in the design of the regulations. Implementers claimed central government could have understood the business context better, and in doing so could have delivered a better policy.

*Government doesn't understand the out-of-home sector and how it works, how store layout is, how consumers work in those environments, they don't understand that at all, which doesn't help. On top of that, you have an industry that is very, very different... So the same tactic can't necessarily be applied to all of those* (Implementer 04).

The diversity of the out-of-home sector also contrasted with a perceived homogeneity of the grocery retail sector. Business participants suggested any lessons by Central Government from regulating supermarkets did not apply to the out-of-home sector.

*You have a number of different supermarkets but essentially it's the same model and way of getting food into your business… whereas the hospitality sector is much more diverse. Just taking, well, this works in a supermarket, let's stick it wholesale onto out-of-home, be it, on allergens, be it on calories or whatever... it doesn't work as simply as that* (Implementer 02).

## Businesses emphasize customer choice and want a level playing field with other businesses

Despite indicating that the OOH food sector may not represent a supportive context for calorie labelling, participants also reported that their businesses wanted to be seen as responsive to customers' needs and expectations.

*We always want to be seen as a company that listens, reacts and hopefully makes our offers open and viable to all parties without losing that reason that they're coming to us for a great occasion… there will be some dishes over 2,000 calories but there'll be a balance across the menus that give us that ability to be able to service everyone's needs"* (Implementer 09).

Although implementers agreed they could support healthier eating, several businesses expressed concerns about a level playing field, both from their own perspective and from the perspective of customers. Issues included larger businesses facing a competitive disadvantage compared to smaller businesses serving similar food that were not in scope of the regulations.

*If you look at a company like [online food delivery company], they would tell you that 50–60% of business they have in their books are actually very small business that don't need to provide that information. (Implementer 11)*

Some implementers suggested smaller out-of-home food establishments were the real public health problem if people ate there on a regular basis.

*So, the chip shop that is serving a little village you know three times a week doesn't have to do any of this, but actually they've got a far greater impact on their local population than a [chain restaurant] or a pub. (*Implementer 06)

## Regulatory context – central government and local authorities

**Structure of enforcement.** Our interviews revealed a decentralised approach to enforcement, whereby Government drafted the policy after consultations with businesses and trade organisations, but policy interpretation was primarily

carried out by businesses partnering with LAs in Primary Authority relationships. LAs were not equal in their capacity to assist with business inquiries due to a combination of insufficient economic and staffing resources. Primary Authorities took an engagement, rather than authoritative, approach when working with businesses (Table 2).

The Primary Authority relationship created enforcement efficiency, whereby business concerns were raised with their Primary Authority, guidance was given, and businesses implemented that advice on a case-by-case basis. However, this decentralised approach to enforcement relied heavily on trust: with LA's trusting that businesses would correctly implement their advice. We term this assumption that businesses were complying in good faith 'presumed compliance' (Enforcer 09), meaning that checks were probably not required because of few complaints from the public (Table 2).

On the other side of the relationship, implementers were sceptical that LAs had the capacity to check for and verify calorie information. Instead of fearing enforcement action, some implementers expressed that customer trust and protecting their reputation were guiding values for implementing the regulations correctly.

*I think the Councils [LAs] aren't even checking. They don't have the budget to sample anything, that is for sure, so you do the due diligence yourself, and I guess, because you are a big brand, you don't want to lose your customers' trust.* (Implementer 04)

Enforcement therefore relied on two layers of trust: LAs trusting businesses to comply because they knew that this compliance helped businesses to maintain customer trust. However, the decentralised enforcement structure sometimes generated confusion as to who had the final word on interpreting the regulations. It was suggested that more guidance from Central Government and direct engagement with businesses could increase confidence in interpreting the regulations (Table 2). More central guidance and support could also help reduce inequalities in capacity between LAs.

*I suppose for us it's training of officers, making sure that we're aware of the requirements and give officers training so that we deal with things consistently within our borough but I think it's also important that training of officers nationally is available so that we all have been trained to the same standard and to offer a consistency across the country as well because a lot of these larger companies will have multiple outlets won't they? (Enforcer 01)*

**Enforcement actions.** Interviews revealed little enforcement activity by LAs for several reasons. LAs rated calorie labelling checks as low priority compared to enforcement activity around more 'acute' food threats – such as allergen labelling.

*Where does it sit on my priorities? Well, the answer's got to be it's somewhere near the bottom because at the end of the day my priority is, is that, what's the risk? … Is that business in a position where if they do something wrong around this piece of legislation they are likely to kill somebody today? (Enforcer 02)*

LAs were not provided with additional resources to support enforcement, so few LAs had the capacity to conduct checks beyond their typical schedule. One enforcer explained that testing places the cost burden on LAs (and ultimately local taxpayers), many of which already have budgetary constraints.

*The enforcement can be done because the [Trading Standards Officers] or [Environmental Health Officers] can go out and they can sample products... that's a cost to the consumers that live in those local authorities if the agencies decide to go out and do testing (Enforcer 06)*

In addition to the limited training and support for enforcement, many LAs had a long backlog of site visits since the COVID-19 pandemic, and in general, out-of-home food business inspections were conducted according to risk assessment criteria.

Large out-of-home food businesses in scope of the calorie labelling regulations were generally considered lower risk relative to smaller food businesses that have less tightly regulated supply chains and food preparation methods.

*We are encouraged as enforcers to visit anything that we categorise as high-risk and anything that is on our system as a new, unrated business. …. Obviously none of those are going to have 250 employees* (Enforcer 02)

## Business impacts

**Implementation process.** Implementers highlighted several impacts on businesses of responding to the regulations. Businesses worked together and with their trade organisations to interpret the regulations. There was some variation in how businesses reported interactions with Central Government, ranging from appreciative of engagement to frustrated with lack of definitive answers. A key criticism of the consultation and subsequent delivery process by implementers was that businesses felt the timeline was too short to develop and implement appropriate delivery systems.

*We were given a year from publication to legislation, however … less than six months from publication of guidance, even when the guidance was published there was a number of questions that [still] needed clarity …and that took several months to obtain. So essentially by the time we obtained all the relevant information we only had really three months…and I think going forward you know, a period of maybe 18 months (Implementer 07)*

Suggestions for improvement included clear guidance for all LAs and businesses, frequently asked question documents, longer timelines to implementation and better stakeholder engagement (Table 2).

*Timing, clarity, detail, engage with your stakeholders. We asked for "can you not do a frequently asked questions?" which is what they did when the gluten regs came in… and that was really helpful.* (Implementer 05)

In contrast, others commended Central Government for engaging in consultations and felt the process went smoothly.

*[Central Government] were very positive as well in their willingness to interact and I have to say … they turned up week after week... their willingness to talk to us was admirable so all of those positive engagements were definitely a good thing that came out of it. (Implementer 04)*

There was also variation in the degree to which implementers expressed that calorie labels were costly for businesses, with some implementers suggesting it created substantial resources burden and others that it was a small addition to typical menu preparations (Table 2).

**Ensuring label accuracy.** Facilitators of implementation included the staff and financial resources necessary to implement the systems that generated and updated accurate calorie information. These included large online databases that could be updated when menus changed. Businesses that handled the transition effectively had sophisticated online data linkage systems that allowed for real-time customization of products.

There were several practical challenges reported to ensuring calorie label accuracy, with customisations being one of the most common. This was particularly the case when component parts (e.g., milk and coffee) add up to a single product of fixed size and the calorie content of each component becomes interdependent on other components (e.g., more milk means less coffee).

*If you have a pizza, you have a Marguerita base, first you have several diameters but, different values for that diameter, but depending on the quantity of additional toppings that you add, it's not as easy as saying, plus 33 calories for mushrooms and 200 and whatever for macaroni because the ratios of those vary... that became really complicated (Implementer 11)*

**Reformulation and menu changes.** Some implementers reported their business had a reformulation or menu change strategy that was either affected by the calorie labelling policy or part of a longer-term nutrition strategy. Reasons for menu changes included avoiding potential embarrassment, particularly through media coverage of extremely high calorie items, and increasing the number of low-calorie options to provide more "choice" for customers.

*Where there was some... potential embarrassment we took those dishes off the menu. We have got dishes on menus that are above the recommended intake of calories for an adult but [at] the end of the day we have choice on our menus and different people need different calorie intake… it's down to their choice. (Implementer 05)*

Some implementers claimed already having nutrition strategies to reformulate products, reduce calories, or to display calories. In these cases, the regulations may have acted as a catalyst for expedited implementation of calorie labelling. While some implementers preferred a gradual, imperceptible reformulation approach ("health by stealth", Implementer 10), others expressed reluctance to reduce calories unless prompted by customer feedback.

*There's been no deliberate sort of take a slice of cheese off just to reduce the calories, but if we were getting feedback that people didn't like the extra slice of cheese then we would take it out for that reason which would then have a knock-on benefit to the calories. (Implementer 06)*

### Impact on customers

Both implementers and enforcers reported receiving few complaints from the public in relation to calorie labelling and expressed uncertainty and scepticism that the regulations impacted consumer purchases. This scepticism related to a number of issues: customers were perceived to be uninterested in calorie information; the wider food environment was considered too unhealthy for a single policy to make a difference; eating habits were considered too ingrained to be responsive to calorie labelling; focusing on calories alone was felt to be a simplistic response to unhealthy eating and the policy was not considered well targeted to those people who most need to change.

Both implementers and enforcers felt a key barrier to policy success was the lack of messaging to the customer about how to use calorie labels. In some cases, the existing contextual messaging that 'adults need around 2000 kcal a day' was considered insufficient (Table 2).

Both implementers and enforcers expressed that the ultimate success or failure of the policy depends on whether people pay attention to the policy and use the information.

*Has it been successful or not, I think time will tell and I think it goes back to my point around the number of people that are actually looking at it (Implementer 08)*

Some implementers suggested a focus on calories as the only nutritional information provided in labels was overly simplistic and may not necessarily direct customers to healthier choices.

*I think if the Government wanted people to know that information, putting calories on something doesn't really tell the customer much about it... those calories could be mostly sugar, those could be mostly fat. (Implementer 03)*

## Discussion

### Summary of main findings

This is the first qualitative investigation of the process of mandatory calorie labelling implementation in England's out-of-home sector. In-depth interviews with individuals responsible for implementing and enforcing the calorie labelling regulations revealed that the potential impact of the policy is influenced by five themes: three contexts, including economic,

regulatory and business contexts, and two areas of impact: businesses and customers. Implementers faced extra demands during a financially difficult time. Delivery and enforcement were decentralized, with varied assistance from LAs due to resource constraints. Enforcement activity was low due to resource constraints, few complaints from the public about missing or incorrect labels, and calorie labelling being deemed low priority by LAs that were more concerned about acute threats like allergens. Primary authority relationships played a key role in resolving businesses' queries about the regulations, creating a sense of "presumed compliance". Businesses complied with regulations to protect their reputation and maintain customers' trust. Both sets of participants were supportive of calorie labelling but believed it would have little impact on customer behaviour. There was some indication that calorie labelling may have triggered or accelerated reformulation of menu items.

## Strengths and limitations

A key strength of this study was the diverse sample of interview participants working in either implementation or enforcement, which provided a broad understanding of the various perspectives on the calorie labelling regulations. Methodological rigor was maintained through several approaches. Credibility was ensured via triangulation, with two researchers (MW and JA) independently coding 15% of transcripts and reviewing the findings with the primary researcher (ME) to confirm consistency. Transferability was enhanced by sampling participants from different regions for enforcers and across various business types for implementers, ensuring a range of experiences was captured. Recruitment saturation was considered as a balance between pragmatism and methodological rigor throughout the recruitment process, and by the study's conclusion there was substantial repetition in topics covered by participants. Dependability was supported by a structured data analysis process using the Framework Method, with consistent coding practices and the use of coding memos to track emerging patterns and reflections. Confirmability was achieved through detailed notes from monthly meetings with the core project team (ME, JA, MW), facilitating regular discussions on coding interpretations and methodological adjustments, and maintaining an audit trail of key analytical decisions to reduce potential bias.

While our study provides insights into the impact of calorie labelling, there are limitations that affect our ability to fully understand its effects on implementation and enforcement, as well as its implications for future policy. We did not conduct interviews with central government policymakers or customers, who would likely have provided useful additional insights. Our specific focus on the regulatory and enforcement structure related to calorie labelling in England may not be generalizable to other country contexts with different economic and social conditions and enforcement structures. Interviews concluded approximately seven months after the policy was implemented, which prevents us from drawing conclusions about longer-term impacts or perceptions of the policy.

## Interpretation and implications of findings

Insights from these interviews have the potential to refine current policies for enhanced impact and contribute to the development of future policies. Although our findings are grouped differently, they encompass key components of process evaluation frameworks such as the Consolidated Framework for Implementation Research (CFIR; [29]). Specifically, they identify how elements—such as intervention characteristics, outer setting pressures, inner setting constraints, characteristics of individuals, and the process of implementation—collectively influenced the adoption, compliance, and perceived effectiveness of calorie labeling in England's out-of-home food sector. We outline each set of implications below according to the analytical themes.

## Economic context

Our findings on the economic context suggest that the cost-of-living crisis may lead some customers to seek value for money instead of using calorie labels to select lower-calorie options. These claims align with other evidence that people with the lowest income quintile in the UK would need to spend half their disposable income to meet Government dietary

guidelines, and that healthier foods are more expensive per calorie [30]. To address the impact of economic factors on food choices, policymakers should prioritize interventions that enhance the affordability of healthier options, and public health campaigns could incorporate messaging that addresses both the health and economic aspects of choosing lower-calorie options. To enhance compliance and optimize the effectiveness of policies, businesses facing financial constraints may benefit from receiving targeted financial or logistical support, thereby increasing the potential benefits to the public.

### Implications for regulation and enforcement

Our findings from the regulatory context suggest additional central guidance could increase efficiencies, particularly in a context of stretched LA resources. Although businesses expressed confidence in their own adherence to protect their reputations, enforcement may still be required to verify these claims. Verifying adherence may also become more challenging if regulations are extended to smaller businesses.

Our study identified two layers of trust in enforcement: LAs trusted businesses to comply with the regulations, while businesses trusted LAs to correctly implement them after receiving guidance. However, other research on the compliance practices of OHFOs in England found 80% of the sampled businesses displayed any calorie information, 67% of business had legible calorie information, and 15% followed all compliance criteria [15]. Those findings suggest that the presumed compliance described in this study may not reflect actual practice. Governments play a key role in holding business entities to account, a role that is diminished if governments lack the resources to enforce their own policies [31]. Although few complaints or enforcement actions were reported, our interviews were completed at seven months post-implementation, and there may be more enforcement contacts in the future once COVID-related inspection backlogs clear. An over-reliance on complaints as an indicator of poor compliance may also lead LAs to underestimate compliance, and additional monitoring may be necessary.

### Business impacts

Areas for policy improvement from a business perspective centred on more guidance from Central Government, for example publishing frequently asked questions. Businesses also criticized the policy for having too short a timeline given the large amount of work required to calculate and accurately display calorie information on menus. However, plans for a calorie labelling policy were announced in a 2018 policy document, years before the April 2022 implementation [32]. Other studies on financial impacts of the calorie labelling policy on businesses would be required to assess the accuracy of cost-related claims.

### Business claims about customers

Although our data reported business claims about customers' beliefs, rather than customers' stated or revealed beliefs, implementers made a consistent set of claims regarding their views of their customers. A commercial determinants of health perspective shows how poor diets are heavily influenced by commercial interests [33], and our interviews surfaced tactics employed by the out-of-home food sector that echo strategies found in a broader industry playbook resisting regulation [34].

One common industry tactic is "to turn a health challenge into a fundamental debate about individual freedom and choice" [35]. Implementers asserted business commitments to providing choices based on customer preferences, even if they were unhealthy. For example, offering some menu items with 2,000 calories "to service everyone's needs" because "different people need different calorie intake" (2,000 calories is the recommended maximum daily intake for an adult). Some implementers agreed their food was not healthy, but that was part of the attraction. This orientation toward customer choice suggests that businesses exist to satisfy customer demand, and that demand is for a treat experience. However, customer preferences are also shaped by industry marketing, which has been shown to promote eating energy-dense, nutritionally poor foods and encourage customer indulgence [36,37]. Implementers also suggested that OOH eating is not

the norm, which could be a strategy to reduce the perceived impact of the out-of-home food sector on population obesity. This evidence suggests that regulating industry marketing practices that promote energy-dense, nutritionally poor foods could complement other regulations of the out-of-home food sector.

There were also several claims from businesses about why they should not be regulated or why calorie labels would not affect customers. Implementers claimed that focusing on calories alone was overly simplistic, may not lead to healthier consumer choices, and the out-of-home sector is exceptional because it is too diverse to regulate as a whole. These claims echo common "complexity arguments" that there is "no one-size fits all" solution used by food, beverage, alcohol and gambling industries to limit the scope of public health interventions and in this case to suggest simple solutions like calorie information are ineffective [34]. Some implementers suggested *other* small out-of-home food establishments were the real public health problem and insisted on the need for a level playing field so all out-of-home eating was subject to the same regulations. This reflects previous claims of 'exceptionalism.'

If implementer claims about the difficulty to effectively regulate are taken at face value, then a potential solution is co-production of the policy among key stakeholders, which could enhance anticipation and mitigation of barriers [38]. However, previous work on public-private partnerships like the Public Health Responsibility Deal found minimal health impact [39], and improving the efficacy of public-private partnerships may require greater monitoring and enforcement resources [40].

Although the doubts expressed by implementers are common industry arguments against regulation, public health policymakers should also pursue policies with a clear evidence-based theory of change [41]. In addition to product reformulation, the other potential mode of action of calorie labelling is via informed consumer decision making. This depends on consumer attention to and utilization of the information provided. Customer noticing and use of calorie labelling policies are generally low across countries. A multi-country study found within jurisdictions with mandatory calorie labeling in restaurants, only 21% of participants noticed and 11% used calorie information [42]. In England, pre-post surveys of approximately 3,000 out-of-home customers found 32% of participants noticed calorie labels post-implementation, and only 22% of those who noticed also used calorie labelling to make their purchasing decision [16]. This study also found no evidence of change in the energy content of purchases [16]. This reflects both implementers' and enforcers' uncertainty that the regulations were likely to impact on consumer behaviour. To address these barriers, messaging strategies could be developed that emphasise social awareness and support for government-led food environment policies [43]. Future menu labelling policies should consider effective label types and designs, such as supportive messaging or alternative labelling (e.g., warning labels), that may have a greater impact on lowering consumer demands compared to information-based policies like calorie labels. Contextual or interpretive labels may help customers select fewer calories [44]. An online randomized controlled trial found added-sugar warning labels on restaurant menus have also led to small reductions in high-sugar menu orders [45]. Although we identified several gaps in compliance and enforcement, policies that make high demands on consumers may not address the key barriers to healthy eating, regardless of the fidelity of implementation.

Finally, the most common proposed solution given by interviewees to improve diets was the suggestion for more customer education. Shifting responsibility to information, education, and individual choice is a common food industry response to regulation [34]. This was also seen in the context of the Public Health Responsibility deal, where despite a claimed commitment to public health, proposed solutions focused on individual behavior changes such as providing information and were ineffective [39].

### Unanswered questions or future research

Future research could examine the long-term impact of mandatory calorie labelling regulations on food business menus and customer behaviours. Businesses may prefer to make gradual changes to menus, and future work should explore long-term menu changes. However, it is possible these changes would have eventually occurred anyway according

to general business strategies of slow reduction in nutrients of concern. We also identified a general perception of low enforcement activity. Whilst this reflects our own empirical findings on enforcement [15], to our knowledge, there have been no studies testing the accuracy of calorie labelling values. Future work could examine accuracy of calorie labels and whether greater resources for enforcement increases compliance and accuracy. More research on consumer attitudes related to the calorie labelling law in England is also needed, particularly in relation to potentially competing economic concerns. A better understanding of the barriers to consumer change, as well as perspectives from central government officials, could lead to improved future policies or complementary policies that reduce barriers to change.

## Conclusions

This qualitative study identified barriers to the effectiveness of calorie labelling regulations, offering insights to guide future policies. Findings highlight the importance of central guidance, adherence verification, and adequate enforcement resources. Policy refinement should consider the economic context, customer expectations, and industry challenges, as well as the type of labelling approaches that optimize impact. Interviews also revealed common industry arguments against the policy's implementation. Limitations include the absence of direct perspectives from customers and central government policymakers. Future research could explore customer engagement with labels to assess long-term impacts.

## Supporting information

**S1 File. COREQ checklist.** A 32-item checklist for reporting qualitative research from interviews and focus groups.
(DOCX)

**S2 File. Participant Information Sheet.**
(DOCX)

**S3 File. Interview topic guide for implementers.**
(DOCX)

**S4 File. Interview topic guide for enforcers.**
(DOCX)

## Author contributions

**Conceptualization:** Tom Bishop, Thomas Burgoine, Andrew Jones, Eric Robinson, Richard Smith, Jean Adams, Martin White.

**Data curation:** Michael Essman.

**Formal analysis:** Michael Essman, Jean Adams, Martin White.

**Funding acquisition:** Tom Bishop, Thomas Burgoine, Andrew Jones, Eric Robinson, Richard Smith, Jean Adams, Martin White.

**Investigation:** Michael Essman, Jean Adams, Martin White.

**Methodology:** Michael Essman, Jean Adams, Martin White.

**Project administration:** Eric Robinson, Jean Adams, Martin White.

**Supervision:** Jean Adams, Martin White.

**Writing – original draft:** Michael Essman.

**Writing – review & editing:** Michael Essman, Tom Bishop, Thomas Burgoine, Andrew Jones, Megan Polden, Eric Robinson, Richard Smith, Jean Adams, Martin White.

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
