## [Decision Letter · Decision Letter 0]

PONE-D-24-11664Implementation and enforcement of mandatory calorie labelling regulations for the out-of-home sector in England: qualitative study of the experiences of business implementers and regulatory enforcersPLOS ONE

Dear Dr. Essman,

Thank you for submitting your manuscript to PLOS ONE. After careful consideration, we feel that it has merit but does not fully meet PLOS ONE’s publication criteria as it currently stands. Therefore, we invite you to submit a revised version of the manuscript that addresses the points raised during the review process.

Authors need to address the queries raised by both the reviewers especially by the Reviewer-2

We look forward to receiving your revised manuscript.

Kind regards,

Muhammad Khalid Bashir, PhD

Academic Editor

PLOS ONE

Journal Requirements:

Additional Editor Comments :

Authors need to address the queries raised by both the reviewers especially by the Reviewer-2

Reviewers' comments:

Reviewer's Responses to Questions

**Comments to the Author**

1. Is the manuscript technically sound, and do the data support the conclusions?

Reviewer #1: Partly

Reviewer #2: Partly

2. Has the statistical analysis been performed appropriately and rigorously? 

Reviewer #1: N/A

Reviewer #2: No

3. Have the authors made all data underlying the findings in their manuscript fully available?

Reviewer #1: Yes

Reviewer #2: Yes

4. Is the manuscript presented in an intelligible fashion and written in standard English?

Reviewer #1: Yes

Reviewer #2: Yes

5. Review Comments to the Author

Reviewer #1: Thank you for giving me a chance to review this paper. This is a good study but there are few comments that should be addressed as listed below:

1. Please provide some information about the interviewer, qualification and any experience in conducting interviews.

2. Please provide some information about data collection tool. How did you develop the semi structured questionnaire? How many sections had and did you do a pilot study before suing it?

3. Did you use a semi-structured interview for both groups? Or you developed for each separately?

4. Please provide information on the study rigor to trustworthiness. How about credibility, transferability, dependability and confirmability. Please explain how did you address them?

Reviewer #2: The study in hand takes up a pertinent issue in a suitable manner. Thy study provides justification for evaluating the role of caloric labelling regulations for the out of home sector in England. This is qualitative study which needs to be supported by suitable theoretical background.

6. PLOS authors have the option to publish the peer review history of their article (what does this mean?). If published, this will include your full peer review and any attached files.

Reviewer #1: **Yes: **Masoud Mohammadnezhad

Reviewer #2: No

---

## [Author Response · Author response to Decision Letter 1]

30 Dec 2024

Thank you for the helpful comments and for the opportunity to revise our manuscript. We have made revisions according to the suggested edits and believe the manuscript is improved for your consideration at PLOS ONE.

Below is a point-by-point response to each request.

Journal Requirements:

Our response: Thank you for this comment. We have formatted the manuscript according to the guidance, and changes are captured as tracked changes. These changes include updating the format for affiliations; updated the format of headings to Level 1, Level 2, and Level 3 headings using 18pt, 16pt, and 14pt font size, respectively; updated headings case to sentence case; table title formatting; adding a section on Supporting Information with correct file names. We also updated the citation style to Vancouver.

Our response: Thank you for this comment. Data supporting this study cannot be made available due to ethical restrictions imposed by the University of Cambridge Humanities and Social Sciences Research Ethics Committee. The interview transcripts contain potentially identifiable and sensitive information. Participants did not consent to the sharing of their data publicly. Given the sensitive nature of the topics discussed and the participants involved, sharing these transcripts, even in an anonymized form, risks compromising participant anonymity. To maintain the confidentiality expected by participants, data access is not possible without significant redactions that would render the transcripts of limited value for secondary analysis. Our primary responsibility is to protect our participants, who provided information in good faith under the assurance of confidentiality.

Additionally, we believe that strict replicability is not applicable in qualitative studies in the same way as quantitative research. In interpretive research, analyses vary significantly depending on theoretical frameworks and researcher interpretation. Sharing heavily redacted transcripts would not contribute meaningfully to replicability or further research.

Additional Editor Comments :

Authors need to address the queries raised by both the reviewers especially by the Reviewer-2

Reviewers' comments:

Reviewer's Responses to Questions

Comments to the Author

Reviewer #1: Thank you for giving me a chance to review this paper. This is a good study but there are few comments that should be addressed as listed below:

1. Please provide some information about the interviewer, qualification and any experience in conducting interviews.

Our response: Thank you for this comment. First author ME is an early career researcher who completed the “Planning and Designing a Qualitative Study” research training conducted by the Social Research Association. He was also guided in monthly meetings with project lead MW, who has published extensively in policy-relevant qualitative research. These characteristics of the interviewer are described in the reflexivity statement at the end of the Methods.

2. Please provide some information about data collection tool. How did you develop the semi structured questionnaire? How many sections had and did you do a pilot study before using it?

Our response: Thank you for this comment. The semi-structured questionnaire (interview topic guide) was developed collaboratively in the research team in response to the research questions. Team members MW and JA have significant experience publishing qualitative research. The interview topic guide was piloted in two instances: first, with a colleague who has been a first author in policy-relevant qualitative interview research (1), and second as a presentation to the Qualitative Research Group, a monthly meeting among qualitative researchers working at the MRC Epidemiology Unit. We do not cite this reference in the paper as it would not be appropriate to cite for substantive reasons. We agree with the reviewer that it is important to discuss how the interview topic guide was presented. However, we also emphasise that the purpose of the interview topic guide was to gather data related to research questions related to implementation and enforcement of this particular policy. We have added details on questionnaire development to the “Data collection: interviews” section of the

Methods.

Reference

1) Jones CP, Forde H, Penney TL, et al. Industry views of the UK Soft Drinks Industry Levy: a thematic analysis of elite interviews with food and drink industry professionals, 2018–2020. BMJ Open 2023;13:e072223. doi: 10.1136/bmjopen-2023-072223

3. Did you use a semi-structured interview for both groups? Or you developed for each separately?

Our response: Yes, we used tailored semi-structured interview guides for each group—implementers and enforcers. While both guides shared broad topics, they were specifically adapted to address the unique perspectives and experiences of each group. The guide for implementers included questions focused on their internal processes, such as how they implemented the labelling requirements, challenges faced, and changes in business practices (e.g., reformulation, customer feedback). In contrast, the guide for enforcers was designed to explore regulatory aspects, including enforcement challenges, compliance monitoring, and interactions with businesses (e.g., verifying label accuracy, assessing businesses' understanding of the regulations). This tailored approach ensured that we captured a range of experiences related to both implementing and enforcing the calorie labelling policy.

To make this clear, we have added more detail on the development of each interview topic guide in the section “Data collection: interviews” within the Methods section.

4. Please provide information on the study rigor to trustworthiness. How about credibility, transferability, dependability and confirmability. Please explain how did you address them?

Our response: Thank you for this comment regarding study rigor. We have added discussion of credibility, transferability, dependability and confirmability to the strengths and limitations section of the Discussion. The use of the Consolidated Criteria for Reporting Qualitative Research (COREQ) checklist also serves as a standardized tool to ensure comprehensive and transparent reporting of qualitative research, which is written in the Methods but not duplicated in the Discussion section to reduce total word count.

Reviewer #2: The study in hand takes up a pertinent issue in a suitable manner. Thy study provides justification for evaluating the role of caloric labelling regulations for the out of home sector in England. This is qualitative study which needs to be supported by suitable theoretical background.

Reviewer #2:

Title: Implementation and enforcement of mandatory calorie labelling regulations for the out of-home sector in England: qualitative study of the experiences of business implementers and regulatory enforcers

Manuscript no: PONE-D-24-11664

Overall Comments:

The manuscript describes the importance of caloric labelling and its implementation in food outlets in England. Barriers and facilitators of this policy impacted in different way. I would like to acknowledge the authors for their valuable contributions in this manuscript. The topic is both timely and important, and the study offers interesting insights and makes its worth impact.

Our response: Thank you for your comments regarding the relevance of this study. Our aim is to contribute meaningful insights into the facilitators and barriers associated with this policy, providing a foundation for future policy development and evaluation.

Suggestions:

Abstract: In the abstract it needs to clearly describe the purpose of the study. How sampling technique suits your methodology. I could not understand how many is your target population and how much is your sample size. What is the ratio between target population and sample size? Either your sample size is statically significant?

Our response: Thank you for your feedback. We have changed the final sentence of the Background section of the abstract to, “As part of a process evaluation, we aimed to examine the implementation of calorie labeling in England, focusing on business experiences and local authority enforcement to identify barriers and facilitators to achieving policy goals”. As this is a qualitative study, our purpose was to gain in-depth insights from large food businesses and local authorities involved in implementing and enforcing calorie labelling regulations, rather than aiming for statistical generalisability. The sample was purposively selected to capture a range of experiences from both business implementers and local authority enforcers across different regions of England. In qualitative research, statistical significance is not applicable because we are not testing hypotheses or seeking to generalize findings to a broader population. Instead, we aimed for theoretical saturation—the point at which additional interviews no longer revealed new themes or insights. This was achieved after 20 interviews, with consistent themes emerging across participants, particularly among enforcers, where limited enforcement activity was consistently observed. It has also been demonstrated empirically that topic saturation is typically achieved within 9-17 interviews (2,3), and meta-themes emerge within 20-40 interviews (4). We found consistent topics across our study sample, particularly among the enforcers, where there was consistently limited enforcement activity occurring.

If we were to calculate the entire target population, previous work found 256 chains were likely to be subject to the calorie labelling regulations when examined in November 2021 (5). If we combine that target population with 1 person at each local authority in England (of which there are 317), then the total target population could be considered approximately 573.

References

2. Hennink M, Kaiser BN (2022). Sample sizes for saturation in qualitative research: A systematic review of empirical tests. Social Science & Medicine. Volume 292,114523. ISSN 0277-9536. https://doi.org/10.1016/j.socscimed.2021.114523.

3. Guest G, Bunce A, & Johnson L. (2006). How many interviews are enough?: An experiment with data saturation and variability. Field Methods, 18(1), 59. 82. https://doi.org/10.1177/1525822X05279903.

4. Hagaman AK, & Wutich A. (2017). How many interviews are enough to identify metathemes in multisited and cross-cultural research? Another perspective on Guest, Bunce, and Johnson’s (2006) landmark study. Field Methods, 29(1), 23-41. https://doi.org/10.1177/1525822X16640447.

5. Huang Y, Burgoine T, Essman M, Theis DRZ, Bishop TRP, Adams J. Monitoring the Nutrient Composition of Food Prepared Out-of-Home in the United Kingdom: Database Development and Case Study. JMIR Public Health Surveill. 2022 Sep 1;8(9).

Introduction: The introduction should clearly state the problem being explored. This introduction highlights the importance of labelling policy but does not clearly describe the source i.e which institution or lab can do the test.

Our response: Thank you for your comment. We have added to the second paragraph of the Introduction that businesses have options for determining calorie content, including laboratory analysis or estimations based on food composition databases and ingredients. This flexibility allows businesses to choose an approach that aligns with their resources. We have also added to the second to last paragraph of the introduction that in light of other findings of imperfect implementation, this study sought to identify the reasons, “This study sought to understand the implementation process and to identify causes of imperfect implementation.”

From Author point of view does labelling mean Nutrition value test of the product and its result?

Our response: In this study, calorie labelling refers specifically to displaying a written statement of energy content (kilocalories, kcal) of menu items, not a full nutritional analysis. Calorie information can be derived through laboratory analysis or estimated based on standard ingredient and portion data. We have added to the second paragraph of the Introduction that the calorie labels must have written labels and a reference statement, thus clarifying the meaning of menu labelling in the context of this study.

Lack of clear research emphasis makes it less likely for the reader to understand the real purpose of the study. Please explain in detail.

Our response: Thank you for this feedback. We have made slight modifications to the final paragraph of the Introduction to clearly explain the purpose of the study:

As far as we are aware, no previous research has explored barriers and facilitators to successful implementation of calorie labelling policies in England or elsewhere. Therefore, the aims of this study were to examine the experiences and processes of implementing calorie labelling in England from the perspectives of businesses and local authority enforcement, identifying barriers, facilitators, and contextual factors that influence policy effectiveness. Additionally, we aimed to identify potentially unforeseen themes that emerged from interviews.

Data and methodology: The authors have used the term Local Authorities (LAs) many times in the manuscript but I am unable to find what does it mean? Does the study include the customer or user of these products with their point of view about labelling formalities? What they are demanding?

Our response: Thank you for this comment, and we acknowledge there could have been a clearer definition given for local authorities. We have clarified the definition of local authorities in the first paragraph of the Methods section, where we first introduce the LA abbreviation:

“We completed one-to-one, semi-structured qualitative interviews with employees of large food businesses (implementers of the regulations) and employees of local authorities (LAs)—local government bodies—who were enforcers of the regulations. There are 317 LAs in England, with a mean population size of approximately 178,000 per LA.”

Regarding customer’s point of view, we state in the Discussion that a limitation of this study is that we did not interview customers: “We did not conduct interviews with central g

---

## [Decision Letter · Decision Letter 1]

Implementation and enforcement of mandatory calorie labelling regulations for the out-of-home sector in England: qualitative study of the experiences of business implementers and regulatory enforcers

PONE-D-24-11664R1

Dear Dr. Essman,

We’re pleased to inform you that your manuscript has been judged scientifically suitable for publication and will be formally accepted for publication once it meets all outstanding technical requirements.

Kind regards,

Muhammad Khalid Bashir, PhD

Academic Editor

PLOS ONE

Additional Editor Comments (optional):

Reviewers' comments:

Reviewer's Responses to Questions

**Comments to the Author**

1. If the authors have adequately addressed your comments raised in a previous round of review and you feel that this manuscript is now acceptable for publication, you may indicate that here to bypass the “Comments to the Author” section, enter your conflict of interest statement in the “Confidential to Editor” section, and submit your "Accept" recommendation.

Reviewer #2: All comments have been addressed

Reviewer #3: All comments have been addressed

2. Is the manuscript technically sound, and do the data support the conclusions?

Reviewer #2: Yes

Reviewer #3: Yes

3. Has the statistical analysis been performed appropriately and rigorously? 

Reviewer #2: Yes

Reviewer #3: Yes

4. Have the authors made all data underlying the findings in their manuscript fully available?

Reviewer #2: Yes

Reviewer #3: No

5. Is the manuscript presented in an intelligible fashion and written in standard English?

Reviewer #2: (No Response)

Reviewer #3: Yes

6. Review Comments to the Author

Reviewer #2: The author has responded the reviewer comments in suitable manner. Now it is satisfactory, and I truly appreciate all the effort you put into this.

Reviewer #3: The authors have made an excellent effort to address reviewer comments. They have sufficiently improved the quality of the paper. I Recommend publication of this paper.

7. PLOS authors have the option to publish the peer review history of their article (what does this mean?). If published, this will include your full peer review and any attached files.

Reviewer #2: No

Reviewer #3: **Yes: **Pomi Shahbaz

---

## [Editor Report · Acceptance letter]

PONE-D-24-11664R1

PLOS ONE

Dear Dr. Essman,

I'm pleased to inform you that your manuscript has been deemed suitable for publication in PLOS ONE. Congratulations! Your manuscript is now being handed over to our production team.

Kind regards,

on behalf of

Dr. Muhammad Khalid Bashir

Academic Editor

PLOS ONE